Domestic carnivore interactions with wildlife in the Cape Horn Biosphere Reserve, Chile: husbandry and perceptions of impact from a community perspective

Schüttler Elke elkeschuttler@gmail.com 1
Saavedra-Aracena Lorena 1
Jiménez Jaime E. 1 2 3 4
1 Sub-Antarctic Biocultural Conservation Program, Universidad de Magallanes , Punta Arenas , Región de Magallanes y Antártica Chilena , Chile
2 Department of Biological Sciences, University of North Texas , Denton , TX , United States of America
3 Department of Philosophy and Religion, University of North Texas , Denton , TX , United States of America
4 Institute of Ecology and Biodiversity (IEB) , Santiago , Chile
Kramer Donald
Electronic publication date: 2018 Jan 4
Publication date: 2018
Volume: 6
Electronic Location ID: e4124
Received 2017 Jun 2; Accepted 2017 Nov 12
Copyright: ©2018 Schüttler et al.
Copyright year: 2018
Copyright holder: Schüttler et al.
License: This is an open access article distributed under the terms of the Creative Commons Attribution License, which permits unrestricted use, distribution, reproduction and adaptation in any medium and for any purpose provided that it is properly attributed. For attribution, the original author(s), title, publication source (PeerJ) and either DOI or URL of the article must be cited.
License URL: https://creativecommons.org/licenses/by/4.0/

Keywords: Conservation, Domestic cat (Felis catus), Domestic dog (Canis familiaris), Free-roaming, Invasive species, Predation, Protected area, Questionnaires, Sub-Antarctic, Survey

Funding: Chilean National Commission for Scientific and Technological Research (PAI-CONICYT) 79140024 Financial support was provided by the Chilean National Commission for Scientific and Technological Research (PAI-CONICYT, Call 2014, No. 79140024). The funders had no role in study design, data collection and analysis, decision to publish, or preparation of the manuscript.

==============================
Background

Hundreds of millions of domestic carnivores worldwide have diverse positive affiliations with humans, but can provoke serious socio-ecological impacts when free-roaming. Unconfined dogs (Canis familiaris) and cats (Felis catus) interact with wildlife as predators, competitors, and disease-transmitters; their access to wildlife depends on husbandry, perceptions, attitudes, and behaviors of pet owners and non-owners.

Methods

To better understand husbandry and perceptions of impacts by unconfined, domestic carnivores, we administered questionnaires (n = 244) to pet owners and non-owners living in one of the last wilderness areas of the world, the Cape Horn Biosphere Reserve, located in southern Chile. We used descriptive statistics to provide demographic pet and husbandry information, quantify free-roaming dogs and cats, map their sightings in nature, and report experiences and perceptions of the impact of free-roaming dogs and cats on wildlife. We corroborated our results with an analysis of prey remains in dog feces (n = 53). With generalized linear models, we examined which factors (i.e., food provisioning, reproductive state, rural/village households, sex, and size) predicted that owned dogs and cats bring wildlife prey home.

Results

Thirty-one percent of village dogs (n = 121) and 60% of dogs in rural areas (n = 47) roamed freely day and/or night. Free-roaming dog packs were frequently observed (64% of participants) in the wild, including a feral dog population on Navarino Island. Dogs (31 of 168) brought home invasive muskrats (Ondatra zibethicus) and avian prey, and over half of all cats (27 of 51) brought home mainly avian prey. Birds were also the most harassed wildlife category, affected by one third of all dogs and cats. Nevertheless, dog-wildlife conflicts were hardly recognized (<9% of observed conflicts and suspected problems), and only 34% of the participants thought that cats might impact birds. Diet analysis revealed that dogs consumed livestock (64% of 59 prey occurrences), beavers (Castor canadensis, 14%), and birds (10%). The probability that dogs brought prey to owners’ homes was higher in rural locations and with larger dogs. There was also evidence that cats from rural households and with an inadequate food supply brought more prey home than village cats.

Discussion

Although muskrat, beavers, and birds were brought home, harassed, or found in dog feces, free-roaming dogs and, to a lesser extent, cats are perceived predominantly in an anthropogenic context (i.e., as pets) and not as carnivores interacting with wildlife. Therefore, technical and legal measures should be applied to encourage neutering, increase confinement, particularly in rural areas, and stimulate social change via environmental education that draws attention to the possibility and consequences of unconfined pet interaction with wildlife in the southernmost protected forest ecoregion of the globe.

Introduction

In parallel with human population growth, the number of companion animals is constantly increasing. Pet and feral dogs (Canis familiaris) have reached an estimated population of 900 million while cat (Felis catus) populations are estimated at 600 million (O’Brien & Johnson, 2007; Gompper, 2014a), existing on all continents except Antarctica (Hughes et al., 2015). Since their domestication thousands of years ago, domestic dogs have had profound roles in human lives. These include companionship, livestock guarding, rescue, hunting, tourism, service animals, and wildlife management (Hart & Yamamoto, 2017). As dogs are part of a diversity of human cultures, their various roles, husbandry, and people’s attitudes towards them have different implications for human-dog-wildlife interactions (Miller, Ritchie & Weston, 2014).

The potential for interactions with wildlife depends on dogs’ and cats’ husbandry, particularly on their confinement. This ranges from complete mobility restriction, for leashed and/or indoor dogs and cats, to feral domestic carnivores that survive independently of supplemental provisioning from humans (Kays & DeWan, 2004; Vanak & Gompper, 2009). Between these extremes, there exists a range of free-roaming animals that are owned or unowned and are, to some extent, subsidized by humans. As subsidized predators, domestic carnivores can reach higher population densities than wild carnivore populations (Gompper, 2014b), leading to complex socio-ecological consequences.

The impacts of free-roaming subsidized and feral domestic dogs include loss of livestock (Baker et al., 2008; Echegaray & Vilà, 2010), aggression towards humans (Schalamon et al., 2006), disease transmission (Matter & Daniels, 2000), and wildlife interference (reviewed in Young et al., 2011; Hughes & Macdonald, 2013; Doherty et al., 2017). Dogs prey upon (Butler, Du Toit & Bingham, 2004; Manor & Saltz, 2004), compete with (Mitchell & Banks, 2005; Vanak, Thaker & Gompper, 2009), infect (Acosta-Jamett, 2009), and disturb (Silva-Rodríguez, Ortega-Solís & Jiménez, 2010; Silva-Rodriguez & Sieving, 2012) wild animals. Suburban cats are successful small vertebrate predators (Woods, McDonald & Harris , 2003; Loyd et al., 2013). On islands, Medina et al. (2011) demonstrated that feral cats wereresponsible for at least 14% of global avian, mammalian, and reptilian extinctions (see also Nogales et al., 2013). Finally, both dogs and cats may hybridize with their wild relatives (Randi, 2008).

While the biology of domestic carnivore-wildlife interactions is often the focus of research, studies on social dimensions are still in their infancy (Miller, Ritchie & Weston, 2014). Understanding the perceptions of free-roaming pet carnivores is indispensable to inform conservation action, because the causes and solutions to this problem directly depend upon the nature of relationships between humans and their domestic pets (Gramza et al., 2016). Conflicts between dogs/cats and wildlife could thus be minimized by a better understanding of how husbandry, attitudes, and behaviors of pet owners influence dogs and cats in their access to and interaction with wild prey or carnivores, particularly when close to protected areas. For example, dog owners felt more obliged to leash their dogs when they thought their dog would harm beach-nesting birds or people (Williams et al., 2009). Recent studies have shown that a more adequate diet (i.e., commercial or prepared food instead of e.g., household scraps) for dogs and cats decreases the probability of preying upon wild animals (Silva-Rodriguez & Sieving, 2012; Sepúlveda et al., 2014). Additionally, in the absence of biological data or for logistical and cost reasons, a community-based assessment may provide suitable information on the impact of particular species on other species (White et al., 2005).

Here, we focus on understanding the interactions with wildlife by populations of free-roaming dogs and cats in a sensitive conservation area of southern Chile, the Cape Horn Biosphere Reserve, from a community perspective. To date, the only substantial human impacts in the biosphere reserve are biological invasions, particularly of wild and domestic exotic mammals that outnumber their native counterparts (Anderson et al., 2006). Additionally, native terrestrial predators are absent on many of the islands of the reserve, including Navarino Island, where the only village of the reserve is located (Anderson et al., 2006). Thus, introduced predators may cause extensive population reductions on native and often naïve prey (discussed in Sih et al., 2010). Although free-roaming and feral dogs are commonly observed by locals on Navarino Island, there are few observations of their impacts. Dogs have been reported to prey upon the southernmost population of guanacos (Lama guanicoe), which is virtually unstudied and considered in danger of local extinction (Cunazza, 1991; González, 2005). Evidence also exists of dogs preying upon nests of solitary nesting waterfowl, such as the Flightless Steamer Duck (Tachyeres pteneres), a species endemic to Patagonia, and nesting colonies of the South American Tern (Sterna hirundinacea, Schüttler et al., 2009). There are no published accounts of local cat impacts.

We administered questionnaires to pet owners and non-owners to address the following objectives: (1) provide demographic and husbandry information relevant for future dog and cat management in one of the last wilderness areas of the globe, (2) quantify free-roaming dogs and cats and map their locations in nature, (3) examine experiences and perceptions of the impact of free-roaming dogs and cats on wildlife and corroborate those with an analysis of prey remains of dog feces, and (4) predict which factors best explain when owned dogs and cats bring wildlife prey home. With regard to the paucity of ecological data on domestic carnivore-wildlife interactions in this sub-Antarctic wilderness area, this study allows a first insight into this conservation problem.

Materials & Methods

Ethics statement

We obtained prior informed consent from each participant by reading a printed statement explaining the project aims, the lack of risks in participating, the possibility of omitting questions, information about use of and access to the results, and that the interview was anonymous and voluntary. The participants agreed to participate by signing a copy of the informed consent; they were also given a copy. Paper and digital questionnaires were stored anonymously. The Scientific Ethical Committee of the University of Magallanes, Chile, certified ethical approval of the instrument.

Figure 1 Map of the Cape Horn Biosphere Reserve, southern Chile.

The Alberto de Agostini and Cape Horn National Parks are core areas of the CHBR and Yendegaia (in light grey) is a recently-created national park. The only large settlement (2,800 inhabitants) is Puerto Williams, on Navarino Island. Eleven Chilean Navy posts are located throughout the reserve on Tierra del Fuego Island (n = 2), Navarino Island (n = 2), and uninhabited islands (n = 7); all are only accessible via maritime transport, except the western Navy post on Navarino Island.

Study area

We carried out this study in the Cape Horn Biosphere Reserve (CHBR) (19,172 km2 terrestrial surface), Chile, located at the extreme southern tip of South America (Fig. 1). The biosphere reserve exists within the Magellanic Sub-Antarctic forest ecoregion, one of the remaining 24 wilderness areas of the world (Mittermeier et al., 2003). The dominant habitats within this ecoregion are unfragmented evergreen and deciduous southern beech (Nothofagus spp.) and Winter’s bark (Drimys winteri) forests, Magellanic peat bogs (predominantly Sphagnum spp.), high-Andean habitats, and glaciers (Pisano, 1977). The human population in the CHBR is of mixed cultural and ethnic origin (i.e., Yaghan indigenous people, Chilean Navy, fishermen, and Chilean and foreign short- and long-term settlers) and is concentrated in the only town within the reserve, Puerto Williams (2,800 inhabitants) on Navarino Island. A small fishing village, Puerto Toro, exists on the eastern coast of Navarino as well as eight rural farm settlements. In the remainder of the biosphere reserve, there are only 11 Chilean Navy posts operated by a new family each year, and one farm on Hoste Island (Fig. 1). The principal economic activities on Navarino Island include fishing, tourism, and small-scale livestock farming. The infrastructure is limited to a dirt road along the northern coast of Navarino; public maritime transport within the reserve does not exist.

Survey

From May 2015 to April 2016, we interviewed 215 households in Puerto Williams, seven farm owners on Navarino Island, and 22 Chilean Navy families living for one year on the 11 Navy posts on different islands in the CHBR (n = 244 total interviews). To test the questionnaire design and adapt the questions, we conducted a pilot study with four trial informants who were later excluded from the dataset.

We calculated a representative 95% probability sample of 215 interviewees in Puerto Williams based on a census of households conducted by ES in May 2015 (490 houses). We used a confidence interval of 5% and applied the finite population correction for smaller populations (Bernard, 2006: 183). We randomly chose 280 households from a map of numbered houses in town (adding 30% to the sample size of 215 to correct for non-responses). When an adult was not present, we left a written message explaining the motivation for our visit and our contact information. We visited each household up to three times before it was replaced. The questionnaires were administered at different times in a face-to-face interview at the participant’s home and took 10–30 min. Two interviewers conducted the interviews in Spanish (n = 92 by ES and n = 152 by LSA).

To meet objective 1, we collected information on dog and cat demographics (i.e., number, age, sex, origin, purpose of pets, number of pups/kittens in previous year, elimination, and temporary and definite pet loss) as well as on pet care/husbandry (i.e., veterinary treatments, food type, in how many households pets eat, and whether owned or unowned street dogs are fed). We asked participants whether they restricted the movements of their dogs (i.e., day and/or night and if free-ranging, why), whether they saw unaccompanied dogs and cats (adults, pups/kittens) outside of town and from where they thought the animals came (objective 2). We asked participants about their observations of problematic dog situations in and out of town, perceptions of possible impacts of free-roaming dogs and cats, and their personal experiences regarding the pet’s interaction(s) with other animals (objective 3). To better understand predictors (i.e., food provisioning, reproductive state, rural/village households, sex, size) of dog and cat interactions with wildlife, we asked owners whether their pets brought wildlife prey home (objective 4). We finished the questionnaire by asking for suggestions for reducing the number of free-roaming dogs, personal data (i.e., age, sex, education, residence time), and the owner’s permission to take a photo of their pet. From this photo, we then classified dog sizes (i.e., small, medium, large) and calculated the mean of two independent estimations by ES and LSA. For farm owners, we added questions on their experience with the loss of domestic animals such as cattle (Bos taurus) or sheep (Ovis aries) to dog attacks. For participants without pets, non-relevant questions were not asked. Some questions were focused only on dogs because dogs produce visible socio-ecological conflicts in the study area.

Dog diet

Dog feces of owned and feral dogs (n = 70) were opportunistically collected within one km of the 11 Navy posts and during trekking events on Navarino Island (June 2015 to April 2016). Each sample was dried, rehydrated, and degreased with detergent, and then grouped into hair, bones, feathers, and rubbish. Using a microscope, we examined the medullary and cuticular patterns of guard hairs to identify up to species level using gelatin as the printing medium (similar to González-Esteban, Villate & Irizar, 1996). As reference collections, we used those provided by the Instituto de la Patagonia, Universidad de Magallanes, complemented by our own additions, and local keys (Chehébar & Martín, 1989). Unfortunately, we could not find a major number of cat feces from different individuals for a sound analysis of cat diet.

Statistics

We first calculated descriptive statistics to analyze basic demographic and husbandry information from the interviews to quantify (i) the number of free-roaming dogs and cats, (ii) problems associated with dogs, (iii) perceptions of the potential impact of free-roaming dogs and cats on wildlife, (iv) occurrence of predation and harassment by dogs and cats, and (v) percentages of different types of prey in dog feces. We then used generalized linear models (GLMs) to examine predictors of interactions with wildlife of owned dogs and cats (Table 1). Our response variable for the dog and the cat model was whether pets brought wildlife home, defined here as any exotic or native wild mammal or bird (PREY). As covariates, we considered diet (dinner leftovers and/or commercial pet food) provided by the animal’s owner (FOOD), sex (SEX), sterility (STERILIZED), owners’ household location (LOCATION, rural or village), and pet size (SIZE, only for dogs). We focused on those covariates as they might identify pet characteristics or levels of care which can then inform recommendations for pet management for the benefit of wildlife. The location was included because rural households are imbedded in wilderness settings and access to wildlife is immediate. Response and predictor variables are explained in detail in Table 2.

Table 1 Candidate models for predicting dog and cat wildlife interactions in southern Chile.

A detailed description of the response and predictor variables is provided in Table 2.

Candidate model sets	Response variable	Quantitative value of response variable	Predictor variables	nb	
Dog model	PREY	19.0% of dogs brought prey home	FOOD+LOCATION+SEX
SIZEa
STERILIZEDa	163	
Cat model	PREY	51.9% of cats brought prey home	FOOD+LOCATION+SEX
STERILIZEDa	52	
Notes.

a Predictor variables not included in final model set due to collinearity.

b We deleted five missing observations from the dog model (3.0%) and one from the cat model (1.9%).

Table 2 Variable description of candidate models for predicting dog and cat wildlife interactions in southern Chile.

The variable description refers to the questionnaire (see Supplemental Information 1).

Variable	Variable type	Variable description and question number	Categories	
PREY	Response	Dog/cat brought wildlife prey home (Q25)	Yes/no	
FOOD	Predictor	Feeding of participant’s dog/cat with leftovers, commercial food/meat, or both (Q15)	0 = leftovers
1 = leftovers and commercial food/meat
2 = commercial food/meat	
LOCATION	Predictor	Dog/cat lives in rural environment (farm /Navy post) or in Puerto Williams	Rural/village	
SEX	Predictor	Dog’s/cat’s sex (Q2)	F = Female, M = Male	
SIZEa	Predictor	Dog’s size (mean of two independent estimations by ES and LSA, Q42)	1 = small, 1.5 = small to medium-sized, 2 = medium-sized, 2.5 = medium-sized to large, 3 = large	
STERILIZED	Predictor	Dog/cat is sterilized (Q11)	0 = not sterilized
1 = pet’s age < 1 year
2 = sterilized	
Notes.

a Variable only used in the dog model, not in the cat model.

As the response variable of the two models was binomial, we fitted generalized linear models (GLMs) with binomial error structure and logit link. The models were parameterized with all possible covariate combinations. Prior to analysis, we explored these data following Zuur, Ieno & Elphick (2010). Collinearity between ordinal covariates was assessed with Spearman correlation coefficients (no coefficients were > —0.4—). We tested the independence of categorical variables using contingency tables (Chi-square and Fisher’s exact tests, p < 0.05). We removed SEX and STERILIZED from the dog model for being significantly associated with LOCATION; similarly, we removed STERILIZED from the cat model for being significantly associated with LOCATION and SEX. FOOD was maintained here, despite its collinearity with LOCATION, to test the same variables in the dog and cat models (the conclusions for models tested with and without FOOD were identical, see Mundry, 2014, for treating collinearity). For model selection, we used Akaike’s Information Criterion corrected for small sample size (AICc). We tested for an interviewer effect by including interviewer as a random effect in the models (generalized linear mixed models, GLMMs), but did not detect any (AIC GLMMs >AIC GLMs of the global models, respectively). We accounted for model selection uncertainty (model weights ωi were <0.9) using full-model averaging (Symonds & Moussalli, 2011). To rank predictor variables in terms of importance, we summed Akaike weights for each model in which the variable under consideration appeared (Burnham & Anderson, 2002). We explored the direction of predictor impacts on the response variable by calculating log odds ratios of the averaged estimates with 95% confidence intervals. Statistical modelling was conducted in R (R Core Team, 2016).

Results

We conducted 215 interviews in Puerto Williams, seven in rural households, and 22 with Navy post families. Only five people in Puerto Williams refused to participate. Of the 244 participants, 61.5% were female, mean participant’s age was 39.5 years (SD 11.6, range 18–76 years), and mean residence time in the biosphere reserve was 11.6 years (SD 14.2, range one month-66 years).

The Puerto Williams participants owned 121 dogs and 36 cats, predominantly for companionship. The seven farm households owned 30 dogs and 15 cats, mainly kept as working dogs and for rodent control, respectively. The 22 Navy families owned 17 dogs primarily for companionship and two cats for rodent control. Two dogs and two cats stayed at the Navy post when families were exchanged after one year (“inherited animals”), the others left with their families. Reproductive control was moderate to high in Puerto Williams (41.7% of dogs and 19.4% of cats not sterilized), but almost absent in rural areas (83.3% dogs, 93.3% cats) and Navy posts (86.7% dogs, 100% cats, but 9 of 17 dogs were ≤1-year, Table 3). Additionally, four participants had eliminated unwanted dog pups.

In Puerto Williams, over half of the dogs and around one third of the cats were vaccinated against rabies (55.4% and 33.3%, respectively) and treated for parasites (60.3% and 36.1%, respectively). Pet owners in rural households did not vaccinate against rabies, but treated for parasites (100% dogs, 40.0% cats). At Navy posts, only dogs were vaccinated and dewormed (64.7%). Pets in Puerto Williams, rural dogs, and dogs at Navy posts were provided mainly with commercial food and/or meat (>77.7%). However, 35 village dogs (28.9%) were fed in more than one household, and 74 interviewees in town (34.4%) reported feeding street dogs on a regular basis (71.8% at least once a week), mostly with leftovers (73.5% of 83 mentioned food items).

Table 3 Demographic dog and cat data, and husbandry results from southern Chile.

We obtained data on owned dog and cat populations via questionnaires from households in Puerto Williams (n = 215), accessible farm households on Navarino Island (n = 7), and Navy families (n = 22, data from two years) living in the 11 Navy posts on different islands within the CHBR.

	Town households (n = 215)	Farm households (n = 7)	Navy posts (n = 22)	
	Dogs	Cats	Dogs	Cats	Dogs	Cats	
Demographic data	
Households with pet ownership (%)	85 (39.5)	30 (14.0)	6 (85.7)	6 (85.7)	19 (86.4)	4 (18.2)	
Mean pet number per household (SD)	1.4 (0.9)	1.2 (0.6)	5 (3.3)	2.5 (2.5)	1.3 (0.5)	1.0 (0.0)	
Total pet number	121	36	30	15	17	2	
Male:female ratio	1.3:1	0.7:1	2:1	0.3:1	0.7:1	0:1	
Mean pet age (SD)	3.7 (3.8)	4.8 (4.1)	3.8 (4.6)	3.0 (3.0)	3.0 (3.7)	3.3 (1.1)	
Number of offspring in previous year	16	0	21	7	9	0	
Local origin (CHBR, %)	66.1	66.7	100	100	29.4	100	
Reproductive control	
Females spayed; males neutered (%)	66.7; 52.2	71.4; 93.3	10.0; 20.0	8.3; 0.0	11.1; 16.7	0.0; 0.0	
Health	
Vaccinated against rabies (%)	55.4	33.3	0.0	0.0	64.7	0.0	
Treated for parasites (%)	60.3	36.1	100.0	40.0	64.7	50.0	
Food provisioning	
Commercial food and/or meat (%)	77.7	94.4	86.7	20.0	82.3	50.0	
Leftovers (%)	12.4	0.0	13.3	33.3	5.9	0.0	
Mix of above (%)	9.9	5.6	0.0	46.7	11.8	50.0	
Dog confinement	
Free-roaming during day or night (%)	30.6	–	46.7	–	82.4	–	
24-h free-roaming (%)	19.0	–	30.0	–	41.2	–	

Sixty-five of 168 dogs (38.7%) roamed freely day or night and 39 dogs (23.2%) were always unrestricted. Using an extrapolated number of dogs for the 490 households in Puerto Williams (n = 275.8 with 1.4 dogs/household), we estimate that 84 dogs (30.6%) roam freely in town during day or night. The most common method of dog restriction (69.4% in town, 53.3% rural, 17.6% Navy posts) was keeping dogs in the house (63.6% of 121 responses), fewer were kept free in the yard (18.2%) or leashed (18.2%). Reasons mentioned for allowing unrestricted movement of dogs in town and rural environments were (i) the owner claimed animal freedom, (ii) leashing may increase aggressiveness, (iii) acclimation to free-roaming and releasing energy, (iv) the dog is not dangerous, and (v) unsuitable facilities (together 77.2% of 70 explanations). Also, 44.0% of 91 non-pet owners thought that street dogs enjoyed their freedom, and most dog owners (87.2% of 86) thought that street dogs roamed into the forest. Finally, 22 of 168 dogs (13.1%) went missing between 12–24 h during the last year (2014/15), among which 13 dogs had disappeared up to one week before returning home. Cats disappeared more frequently (n = 18, 34.0%); 14 cats for 2–7 days. Over the last 10 years, 35 pets (23 dogs, 12 cats) never returned.

Free-roaming dogs not accompanied by people outside of town were frequently observed on Navarino Island (63.9% of 244 participants), whereas cat sightings in the wild were almost absent (6.1%, Fig. 2). The greatest distance of sighted dogs and cats was 19.4 km and 5.2 km from the northern settled coast, respectively. Neither dogs nor cats were seen roaming around Navy posts, except near the two posts on Navarino Island. Dogs were mostly observed in packs, with a median pack size of four dogs (mean 6.6, SD 7.5, range 2–60, n = 172 sightings), while only 9.2% of the sightings were single dogs. Dog pups (abandoned or feral) outside Puerto Williams were sighted by 52 participants (21.3%) with a mean litter size of 4.0 (SD 2.3, range 1–12). Four participants observed pups and kittens (n = 17) abandoned in cardboard boxes outside town. Apparently, the landfill, approximately 500 m from Puerto Williams, was a point of attraction for dogs, as 12 participants observed dog packs with a median size of eight (mean 10.75, SD 8.0, range 1–25, Fig. 2) in this area.

Figure 2 Free-roaming dog and cat sightings on Navarino Island, southern Chile.

Approximate sighting locations of unaccompanied adult dogs and cats, dog pups (abandoned or feral), and kittens (abandoned) from n = 227 sightings by 143 participants during 2014/15. Dog sightings are shown in different classes of pack size.

Over half of the participants (55.9%, n = 222) on Navarino Island had experienced problems with dogs in Puerto Williams during 2009–2015 (83.9% had occurred during the last year, 2014/15), whereas 41 participants (18.5%) reported problems outside the town (61.4% during 2014/2015). Predominant dog problems in town were conflicts with people (biting, attacking, frightening, disease transmitting, accidents, and stealing; 40.6% of 143 problems, 24.1% concerned children) and free-ranging domestic animals (cows, horses, and their offspring) in town, mostly foals (37.1%, Fig. 3). Outside of town, people experienced conflicts between dogs and domestic animals, particularly involving cattle (77.3% of 44 problems), whereas only two people saw dogs feeding on wildfowl eggs (4.5%).

Figure 3 Experienced and suspected problems with dogs and cats in southern Chile.

Problematic experiences with dogs during 2009–2015 (A) inside (n = 143) and (B) outside (n = 44) of Puerto Williams, suspected dog problems (first problem mentioned) (C) inside (n = 221) and (D) outside of town (n = 202), and suspected cat problems (first problem mentioned) (E) outside of town (n = 77). Conflicts with people included biting, attacking, frightening, disease transmitting, accidents, and stealing. Dog-domestic animal problems referred to killing, attacking, or feeding on free-ranging domestic animals such as cows, horses (Equus caballus), sheep, pigs (Sus scrofa), and cats, whereas cat-domestic animal problems referred to preying upon chickens (Gallus gallus domesticus). Conflicts with wildlife included killing wild animals such as birds and their eggs, North American beavers (Castor canadensis), and guanacos, or harming ecosystems. Conflicts with conspecifics were fights among dog packs or between cats, and disease transmission. Dog feces and waste dispersing were considered hygienic problems. “Other” includes dog and cat overpopulation, bad image for tourists, and barking. Images of animals represent predominant animals involved in dog/cat problems.

Beyond personal experiences, most participants thought that free-roaming dogs caused problems both in and outside of town (91.9% and 89.2%, respectively). In town, suspected dog problems mainly involved people (68.8% of 221 problems), whereas outside of town concerns involved domestic animals as well as people (Fig. 3). Dog-wildlife conflicts (e.g., involving guanacos) were only mentioned 19 times (9.4% of 202 problems). However, when asked directly whether feral dogs could have negative impacts on wildlife and whatkind of wildlife could be affected, most participants said yes (80.8%, n = 239) regarding birds (67.3% of 349 problems). Guanacos were only mentioned 16 times among the affected wildlife (4.6%).

Regarding suspected cat problems, only one third of participants (33.8% of 240) associated problems with them outside of town, particularly with cats harassing and eating wild birds and their eggs (67.5% of 77 problems, Fig. 3).

Five of seven farm owners reported losing domestic animals due to unrestricted dogs from Puerto Williams or feral dogs during 2014/15. The estimated total loss during these events were 62 calves, 25 cows, and 30 sheep, while 30 sheep and two calves were injured. The losses on the five farms corresponded to 3.3%, 11.5%, 16.7%, 18.8%, and 35.0% (cows), and 75.0% (sheep) of owned livestock.

Thirty-one village and rural dogs (18.5%) brought prey home, mainly invasive muskrats (32.5%) and birds (27.5%, Fig. 4). Among avian prey were songbirds (Austral Blackbird Curaeus curaeus, Austral Thrush Turdus falcklandii, Rufous-collared Sparrow Zonotrichia capensis), waterbirds (terns, Upland Goose Chloephaga picta), and raptors (Chimango Caracara Milvago chimango). Over one third of all dogs (n = 64) were observed harassing (but not killing) other animals, particularly birds (38.0% of 79 items mentioned: Chimango Caracara, Patagonian Sierra-Finch Phrygilus patagonicus, Upland Goose, and ducks such as Flightless Steamer Ducks), other dogs (16.5%), and horses (15.2%); three dogs harassed native mammals (i.e., foxes and seals).

Figure 4 Preyed and harassed animals by dogs and cats, and suspected feral dog prey in southern Chile.

Prey brought to owners by 31 of 168 dogs (n = 40 mentioned items), animals observed to be harassed by 64 dogs (n = 79 items), prey brought to owners by 27 of 52 cats (n = 33 items), animals observed to be harassed by 19 cats (n = 25 items), and suspected prey of feral dogs (n = 494 items) by 244 participants. “Other” includes fish (dog prey); fish, foxes, and seals (harassed by dogs); bats (cat prey); and bird and horse feces, fish bait, foxes, seals, and vegetable material (suspected feral dog prey). “Small livestock” refers to chickens, pigs, and sheep. All bird species mentioned were native, except for the house sparrow, a species mentioned among the cat prey.

Over half of all village and rural cats (n = 27) brought prey home (birds were 57.6% of 33 prey items): among songbirds, the Austral Thrush, Fired-eyed Diucon Xolmis pyrope, House Sparrow Passer domesticus, Patagonian Sierra-Finch, and among waterbirds, ducks and terns. Birds were also the most commonly harassed prey group (72.0%) by 18 cats (only one species was referred to here: Patagonian Sierra-Finch). The 244 participants mentioned diverse food items they thought feral dogs would eat (Fig. 4). Domestic livestock was the most important group mentioned (39.9%), whereas native birds and guanacos were less recognized (15.2% and 2.2%, respectively).

Of 70 fecal samples from owned and feral dogs in rural zones of the CHBR, we could not identify prey remains in seven, and 10 feces were excluded for only containing dog hair. The subsequent diet analysis revealed that the main food item (64.4% of 59 prey occurrences) found in 53 feces was ungulates (i.e., horses and cows, which could not be distinguished here), followed by beavers (13.6%), birds (10.2%), mice (5.1%), rubbish (5.1%), and Fuegian red fox (Pseudalopex culpaeus, 1.7%).

Three models best explained whether dogs would bring prey home (Table 4). The most important variable with the highest summed Akaike weights (ω, upper limit = 1.0) was LOCATION (ω = 0.99); FOOD (ω = 0.54) and SEX (ω = 0.30) had less importance. The averaged estimates indicated that dogs in rural areas were more likely to bring prey home (Fig. 5A), whereas an adequate diet and the dog’s sex had little influence (their confidence intervals overlapped the odds ratio at one). LOCATION (ω = 0.99) and FOOD (ω = 0.99) were the most important variables in the cat model, whereas sex played a minor role (ω = 0.24, Table 4). Based on the averaged model estimates (Fig. 5B), there is evidence that rural cats that received more leftovers brought more prey home than village cats, whereas the cat’s sex was a poor predictor. From the collinear variables removed before modelling, only SIZE had a significant association with LOCATION and PREY, respectively (Fisher’s exact tests, p < 0.001), indicating that not only rural, but also larger dogs were more prone to prey upon wildlife.

Table 4 Best-ranked generalized linear models for predicting dog and cat wildlife interactions in southern Chile.

Summary of model selection for models with ΔAICc < 2. K indicates the number of parameters per model, ΔAICc distance from lowest AICc, and ωi model weight.

Model set	Competing models	k	AICc	ΔAICc	ωi	
Dog model	FOOD+LOCATION	3	150.72	0.00	0.37	
	LOCATION	2	151.03	0.32	0.32	
	FOOD+LOCATION+SEX	4	152.32	1.60	0.17	
Cat model	FOOD+LOCATION	3	62.19	0.00	0.75	

Figure 5 Model averaged odds ratios for models predicting dog and cat wildlife interactions in southern Chile.

Plots show the model averaged parameter estimates as odds ratios on a log scale with 95% confidence intervals (CI) for the (A) dog model and (B) cat model, where LOCATION (dogs and cats) and FOOD (only cats) best predicted whether pets brought wildlife prey home. The other variables had confidence intervals that clearly overlapped the dashed line at 1, which implies that there is no direction of the parameter estimate.

Importantly, participants suggested several measures to diminish the number of street dogs, including reproductive control (18.9%, n = 534 suggestions), registration (14.6%), responsible pet husbandry (11.2%), establishing an animal shelter (10.5%), education and adoption campaigns (9.2%), controls and penalties (8.4%), movement restriction (8.1%), animal health (7.3%), limiting the number of dogs per family or not abandoning dogs (7.1%), and euthanasia (4.7%).

Discussion

In the absence of biological studies, this survey provides a first understanding of the interactions of domestic carnivores with wildlife in a sub-Antarctic wilderness setting. A representative sample of the local population in the Cape Horn Biosphere Reserve gave insight into pet husbandry and perceptions of impacts of free-roaming dogs and cats. We found that free-roaming dog packs were frequently observed (64% of participants) in natural areas on Navarino Island. Many of these individuals may be owned, as many participants indicated their dogs were not confined (65 of 168 dogs roamed freely day or night, and 22 dogs had even disappeared for 24 h or more). Further, travel distances of free-roaming owned rural dogs vary, with reports of 4 km (Sepúlveda et al., 2015) or up to 8–30 km (Meek, 1999). However, such large foray distances are an exception. Finding a village dog at a distance >1 km from its home had a 10% chance in a study of dogs scavenging sea-turtle nests (Ruiz-Izaguirre et al., 2014), and most rural dogs even spent 95% of their time <200 m from their households (Sepúlveda et al., 2015).

Our findings indicate that there is evidence of a feral dog population on Navarino Island. The participants reported sightings of unaccompanied dog packs in remote parts of the island (up to 19.4 km from the northern settled coast; Fig. 2) and sightings of dog pups (feral or abandoned) outside town (52 participants). They identified 52% of the 172 sightings as feral dogs. This may be an over-estimate, but given that free-roaming village dogs are easily recognized in the small town of Puerto Williams, it is likely that participants could distinguish between owned and feral dogs. However, it is not clear whether this presumably feral population has achieved long-term human independence, as for example the dogs eradicated from Isabela Island, Galápagos (Reponen et al., 2014). The reported population of abandoned dog pups and missing dogs may have been recruited into feral dog packs (e.g., Boitani et al., 2017). The importance of the local landfill (12 sightings of dog packs of 8–11 animals on average) as a food subsidy warrants further investigation, as these novel ecosystems can produce a variety of positive and negative impacts on vertebrate species exploiting them (Boitani et al., 2017; Plaza & Lambertucci, 2017). For cats, the few sightings (5% of participants) in natural areas were along the northern-settled coast, except for one cat sighted 5.2 km south of the coast. Further phenotypical, genetic, and ecological research is needed to better understand the feral dog and possible feral cat population and their impacts on Navarino Island.

Although there were 227 dog sightings during 2014/2015, dog-wildlife conflicts of free-roaming dogs were hardly recognized (4.5% of 44 observed problems; Fig. 3). The direct observation of dog-wildlife interactions is probably a rare situation, particularly with mammals, as the mammalian community on Navarino Island is limited (Anderson et al., 2006). Only five terrestrial native species exist: two species each of bats and mice, and the vulnerable guanaco. Among exotic mammals, there are three elusive wild species (North American beaver, American mink Neovison vison, and muskrat) and free-ranging domestic mammals such as cows, horses, sheep, and pigs. Dog interactions with exotic mammals may not have been considered as true wildlife-conflict by the participants as there was a general consensus in the community about the need for a population reduction of exotic mammals such as mink and beavers (Schüttler, Rozzi & Jax, 2011).

Guanacos have not been sighted along the northern coast for many years (González, Zapata & Marín, 2002), and their densities were as low as 0.14 individuals/km2 on the northeastern coast of Navarino Island during 2002–2005 (González, 2005). Thus, it is almost impossible to see depredation or harassment of guanacos by dogs (one piece of photographic evidence was taken by Denis Chevallay in 2002). However, individual dog attacks on rare species may impact their persistence substantially (e.g., pudus, Silva-Rodriguez & Sieving, 2012; mountain gazelles Gazella gazella, Manor & Saltz, 2004). Therefore, future studies on dog impacts on the southernmost isolated population of guanacos are an urgent need.

The likelihood of interactions among dogs and avian species has the potential to be much higher, as birds, among them many seabirds, are the most diverse and abundant group among vertebrates in the CHBR (Rozzi et al., 2006). Indeed, eleven dogs brought bird prey home, and 30 dogs were observed by their owners to harass birds (songbirds, waterbirds, and raptors; Fig. 4). However, these experiences were not translated into the context of a possible “dog-wildlife” conflict: only 9% of 202 suspected dog problems outside town were dog-wildlife problems (Fig. 3); most were dog-domestic animal (54%) or dog-people conflicts (35%). On the one hand, this may be due to a lack of knowledge of the local fauna by short-term residents; further, Cape Horn biocultural identity is missing in the classrooms. Rozzi et al. (2008) reported an absence of native fauna in the minds of local short-term residents who primarily mentioned cosmopolitan roses and apple trees as local plant species. On the other hand, the absence of dog-wildlife interactions in the participants’ minds might indicate that dogs are mainly perceived as domestic animals that act in a human-dominated context and not as carnivores in a natural ecosystem. Personal safety was also the most common public concern regarding free-roaming dogs and cats in central Italy (Slater et al., 2008). This perception may be attributed to the historical attachment bonding between the dog-human dyad believed to be similar to a child-parent relationship (reviewed in Payne, Bennett & McGreevy, 2015).

For cats, awareness of possible cat-wildlife problems, particularly with birds, was higher (68% of 77 problems; Fig. 3). On the one hand, these problems might be more visible, at least for cat owners, whose cats brought birds home (37% of cats in this study; Fig. 4). This number, however, probably clearly underestimates true capture rates, as Loyd et al. (2013) demonstrated with animal-borne video cameras worn by urban cats in the United States (see also Kays & DeWan, 2004). On the other hand, Arahori et al. (2017) showed that owners’ views of their cats and dogs differed; for example, cat owners had a weaker tendency to regard their pets as family members than dog owners. This perception may also influence their view on how cats behave outside their homes (i.e., as carnivorous species).

Besides prey brought home by dogs and cats, our diet analysis of dog feces showed that dogs indeed preyed upon domestic animals such as cows and horses (64%); but beavers (14%) and birds (10%) were present in feces as well. This finding is consistent with cow losses reported by Navarino’s farm owners. However, part of the diet might also be a result of scavenging animals that died from other causes, such as disease (e.g., Butler & Du Toit, 2002). More biological methods, such as analyzing cat diet or observations of what free-roaming dogs and cats actually do, are needed to further validate and complement the self-reported observations on wildlife-interactions in this survey.

Finally, with GLMs, we showed that rural provenance, large dog size, and an adequately food supply for cats played a significant role as predictors for bringing wildlife prey home. When rural households are imbedded in wilderness settings and spatial and behavioral barriers between domestic and wild animals lacking, the apparent consequence are higher depredation rates of (larger) dogs and cats on wildlife. Increasing the confinement of those pets should thus be part of management strategies. Unlike other studies, an inadequate food supply (i.e., higher percentage of leftovers) was not associated with dogs preying upon wildlife. This may be due to methodological differences. Silva-Rodriguez & Sieving (2011) and Ruiz-Izaguirre et al. (2014) considered body condition score and metabolic energy intake, whereas we only relied on the participants’ statements. To some extent, the social desirability bias (where the participants wish to appear socially or morally worthy, Maccoby & Maccoby, 1954) might underlie these differences by biasing results toward more frequent feeding with a commercial food diet. While restricting boundaries for farm dogs and cats is probably a difficult task due to their roles as working dogs or for rodent control, the necessity of improving nutrition could be more acceptable to owners, not only for lowering wildlife depredation, but also for pet welfare. In general, where pet owners are less susceptible to arguments based on wildlife protection, welfare-based arguments (such as reducing risks of road accidents by confinement) could represent an alternative approach to enhance pet husbandry (Hall et al., 2016).

Conclusions

Unconfined dogs and cats in the Cape Horn Biosphere Reserve interact with wildlife, although this is almost unrecognized by the local community, particularly with dogs. To guarantee the future intactness of this wilderness area, it is essential to put the possible impacts of free-roaming pet carnivores on wildlife into perspective. This should be done using an integrative approach that respects the many dimensions of pet carnivores in their beneficial and problematic interactions with their human, conspecific, and natural environment: (1) although over half of pets in Puerto Williams were sterilized, encouraging neutering, particularly in rural areas, could reduce pet density and avoid elimination or abandonment of unwanted pups/kittens. Moreover, sterilized dogs were described to show lower rates of escaping from home and less roaming behavior (Neilson, Eckstein & Hart, 1997; Spain, Scarlett & Houpt, 2004, but see Garde et al., 2015), which might lower their access to wildlife. (2) Despite its probable unpopularity (e.g., Grayson, Calver & Styles, 2002), we recommend that dog and cat ownership should be completely banned in families living in Navy posts, as they are located within wild insular environments (some in National Parks) where pet interactions with wildlife can be particularly severe (Medina et al., 2011) and pets can become feral. (3) The evidence of a feral population of dogs on Navarino Island (where native predators are absent) urgently needs biological methods of confirmation such as GPS monitoring (Young et al., 2011) and an assessment of its impacts. Feral dogs are still poorly investigated (Boitani et al., 2017), and management implications for feral dogs are challenging (i.e., they may include the removal of dogs) due to their avoidance of human contact (Boitani & Ciucci, 1995) and restriction to natural habitats. (4) Increasing dog and cat confinement is beneficial to prevent not only pet-wildlife interactions, but also many socio-economic problems. In-house/yard confinement largely depends upon cultural settings (Hsu, Severinghaus & Serpell, 2003; Jackman & Rowan, 2007) and is low in many countries in Africa, Asia, and South America (Reece, 2005), including Chile, which has high percentages of free-roaming dogs in rural settings (67%, Acosta-Jamett et al., 2010; 84–91%, Silva-Rodriguez & Sieving, 2012; 92%, Sepúlveda et al., 2014). A new Chilean law promoting responsible pet ownership was recently enacted (Ministry of Health, 2017) and could be a first step towards regulating confinement by national legislation which was considered far from sufficient in 2014 by Bonacic & Abarca (2014). Meanwhile, bells, electronic alarms, or colorful collar covers are an option for free-roaming cats to reduce depredation rates on wildlife (Gordon, Matthaei & Van Heezik, 2010; Calver & Thomas, 2011; Willson, Okunlola & Novak, 2015), as well as advice on environmental enrichment techniques for indoor cats (Rochlitz, 2005), outdoor enclosures, and leash training (Hall et al., 2016). (5) Finally, an instrument such as this survey can reveal what problems with free-roaming pets exist; thus, it provides insight into impacts in the absence of biological studies or complementary to them (e.g., dogs interacting with rare and endemic mammals were only detected through interviews in Silva-Rodriguez & Sieving, 2011). Owners’ and non-owners’ suggestions can also contribute to creating acceptable measures towards responsible pet ownership (Calver et al., 2011). For the CHBR, in addition to the above mentioned technical and legal solutions, social change to improve pet management for wildlife benefits could be stimulated through education about the vulnerability of native fauna to dogs and cats, attractive education material such as puppet videos, documentaries, or animal-borne films for school pupils, and action days such as walking with leashed dogs or breakfasts with dog owners (further examples in Miller, Ritchie & Weston, 2014).

Supplemental Information

Supplemental Information 1 Original questionnaire administered in southern Chile

Questionnaire for dog and cat owners, and non-owners in the Cape Horn Biosphere Reserve, translated from Spanish into English. Questions not relevant for non-owners were not asked.

Click here for additional data file.

This work was only possible due to the willingness and interest of the local community of Puerto Williams, farm owners, and Chilean Navy families to participate in our survey. We are grateful to the Chilean Navy for facilitating maritime transport to the Navy posts on different islands within the CHBR. We would like to thank Eduardo Silva, Ramiro Crego, and Silvia Llanos for their advice in the questionnaire design and Mariela Alarcon and her team from the Scientific Ethical Committee of the University of Magallanes for improving an earlier version of the questionnaire. Amy Wynia, Nancyrose Houston, and two anonymous reviewers made valuable comments on an earlier version of the manuscript. We are also grateful to Andrés Mansilla and Ricardo Rozzi for their support in this study.

Additional Information and Declarations

Competing Interests

Author Contributions

Human Ethics

Data Availability

The authors declare there are no competing interests.

Elke Schüttler conceived and designed the experiments, performed the experiments, analyzed the data, contributed reagents/materials/analysis tools, wrote the paper, prepared figures and/or tables, reviewed drafts of the paper.

Lorena Saavedra-Aracena performed the experiments, prepared figures and/or tables, reviewed drafts of the paper.

Jaime E. Jiménez conceived and designed the experiments, contributed reagents/materials/analysis tools, reviewed drafts of the paper.

The following information was supplied relating to ethical approvals (i.e., approving body and any reference numbers):

The Scientific Ethical Committee of the University of Magallanes, Chile, certified ethical approval of the instrument.

The following information was supplied regarding data availability:

As the raw data for this survey was digitized in Spanish, and some of the answers can be used to ascertain the participants’ identities, the raw data cannot be shared for publication.

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
