# Peer review of "Domestic carnivore interactions with wildlife in the Cape Horn Biosphere Reserve, Chile: husbandry and perceptions of impact from a community perspective"

_PeerJ, doi:10.7717/peerj.4124_

## Round 0.1 · original submission · Major Revisions

This study used a survey of residents near a wilderness protected area in Chile to examine attitudes toward free-roaming dogs and cats, experiences of free-roaming dogs and cats, perceptions of the potential impact of free-ranging dogs and cats on wildlife, and predictors of domestic carnivore access to wildlife (measured by confinement of pets), evidence of conflict with wildlife (measured by frequency of pets bringing prey home), and awareness of conflict with wildlife. Outside of town, free-roaming dogs had been observed by many participants but free-roaming cats had not. Conflicts with wildlife were not frequently raised among problems caused by free-roaming dogs (in comparison to problems with people and domestic animals), but most participants acknowledged that dogs and cats could cause problems for wildlife. Confinement of dogs was not associated with the level of care or attitude toward freedom of movement for dogs but was associated with smaller dog size, access to the house and having a fenced yard. Only about 15% of dogs brought prey home and this was not associated with amount of feeding or sex of dog but was associated with larger size and rural location. Suspecting that dogs feed on wildlife predicted awareness of wildlife conflicts. I did not see a synthetic statement regarding attitudes.

Reviewer 1 indicated that the manuscript was a clear presentation of a well-conducted study but recommended rejection based on its limited nature. This reviewer specifically mentioned the limited number of interviews and geographical area of the study as well as relatively minor or obvious conclusions and vague recommendations, suggesting that it would be more suitable for a local journal.

Reviewer 2, in contrast, found that the knowledge gap and specific objectives were unclear, that the methods were unclear, and that the interpretation required a more critical approach because of the use of a survey method without external validation. This reviewer also recommended rejection but in comments for the editor also suggested possible publication in a more local journal. This reviewer's comments noted several points where the methods require clarification.

It appears that the rejection proposed by both reviewers was related to the limited scope of the study and to problems in communication of the aims and methods rather than to any fundamental flaw of the study or its conclusions. However, PeerJ specifically excludes degree of scientific advance or potential audience size as criteria for publication, and it seems likely that clarification of the aims and methods could be achieved by appropriate revision. Furthermore, many studies can be local in nature but have broader implications for other areas. Therefore, my decision is to recommend major revision to address the problems with the clarity and integration, methods, and broader implications of the study.

Editor's Comments

Organization. I found it difficult to determine the contribution of the manuscript, in part because the integration of objectives, methods and results was relatively weak. Some possible solutions to this problem are the following:
• Consider restating the objectives in terms relating specifically to the methods and results.
• Organize the methods, results, and discussion (and abstract) so that they follow the same order as the objectives and are clearly linked to those objectives.
• When presenting the survey, show which types of questions were intended to accomplish which objectives.
• Supplementary Table S2 seems critical for understanding the study, so should be included in the main article, probably integrated with Table 1. It would be much more helpful for readers if Table S2 was ordered by the model rather the alphabet and that you include all the independent and dependent variables.

Contribution of the study. It is not obvious what precisely this study has contributed to the field.
• Please try to address the broader implications in your Discussion. Although you sometimes refer to differences or similarities between your findings and other studies, it is not easy to determine more broadly what is novel and what reinforces or contradicts previous findings. If your approach is unusual, as implied in the Introduction, you should be able to indicate more explicitly what findings you have contributed to this research field.
• Furthermore, your discussion could address the potential decisions of managers hoping to reduce the impact of dogs and cats roaming in other protected areas. For example, what specific benefits could arise from a survey? Are these adequate to justify the time and expense of carrying it out? How could such surveys be improved as a result of your experience? Have you found anything which suggests potential actions to be tried to reduce wildlife harassment in protected areas? I am not looking for extensive speculation here but making explicit what insights and contributions result from your study.

Other concerns
• The reader should know the quantitative values for all response variables. For example, L209ff details methods of restriction, but the response variable of the model is whether or not restriction occurred, and its value is only mentioned in Table 2. Awareness and bringing prey home variables are not included in this table. Consider emphasizing the value of each response variables in the text.
• You should explain why the models were applied only to dogs?
• I am concerned that some of your findings involve circular reasoning in which the predictor and response variables are logically related. For example, fenced yards and access to the house were negatively associated with free ranging dogs. However, it seems that all 3 variables measure the same thing because people who restrict their dogs need a way to do so. Similarly, isn't the perception that dogs feed on wildlife a component of awareness of dog-wildlife conflict?
• Although English grammar and style are generally quite strong in the manuscript, there are a few problems. I have attached an annotated pdf in which I have highlighted errors in grammar, punctuation or word choice or confusing sentence structure. I have not suggested corrections to all these because the text is likely to change. However, you should be aware of these issues and have one or two native English speakers carefully edit your manuscript before resubmission. Note also some errors of consistent use of capital letters indicated in the references.

Reviewer 1 ·

Basic reporting

The paper shows a clear language throughout. However, I’m not a native English speaker and can’t not assess if it can be improved.
Context and background in intro are well raised and most references are adequate. Structure fits the journal standards. Figures and tables are adequate.

Experimental design

Research and experiment are well designed and performed to ethical standards. Methods are described in detail and data allow replications. Statistical procedures are adequate and especially GLMs well applied, as well as model selection based on AICc ranking.

Validity of the findings

My main concern about the paper is the application and weakness of the results. Sample size is relatively small and geographic ambit restricted. Although data seems robust and statistically significant results seem of limited application. I suggest dealing with this discussing what happens in the rest of the area (Biosphere Reserve). Is the problem with dogs similar in rest of the reserve? Do property and households follow the same pattern than in the island?
The conclusion of the paper is “free-roaming dogs are perceived to interact in an anthropogenic context” and the recommendation is to develop environmental education actions. In most of the places where free-roaming dogs cause problems the solution comes from control actions or culling and always in the origin of the problem is the man hand. What kind of education actions does propose the authors? More details are needed here. Environmental education can be a proposal for the future but does not solve actual problems. Therefore, this kind of guidelines should be complementary of other actions. Which other solutions can be applied there?

Reviewer 2 ·

Basic reporting

Introcution lack of relevant information that help to define the necesity of this research

Experimental design

- Research questions are difficult to understand
- Methods are not well described

Validity of the findings

- Although data obtained from questionaires are used to obtain information regarding impact of domestic animals upon wildlife, this data must be used with caution and be very autocritic with the data obtained if no other confirmation method is used. If valid, data provided in this MS would fit better in a local journal.
- It is not clear enough how model building was carried out biological and stasticaly.

Additional comments

General comments
Ms aimed to understand perceptions of impacts of free-roaming dogs and cats to wildlife. Although the objectives of this MS are of interest in the area, there are several aspects that difficult fulfilling the objectives proposed in this study. Overall, it’s not clear for this reviewer what authors wanted to test with the questionnaire and if data they could get with this could be used for what they wanted to predict, which difficult interpretation.
In introduction it’s is not clear what was the aim of the study. 1) it’s about how free-ranging dogs and cats can get access to a conservation area? Can that be assessed through questionnaires??, The questionnaire can give perceptions more that access to a conservation area. Also, questions are applied to determine predictor of dog’s access to wildlife, which is not clear how questions asked can help to explain this. Methods are not clear and result section is not well written. The MS is about domestic carnivores and in some paragraph cats are included, models run, but in others only dogs are mentioned.

Specific comments
Introduction
L68-69: Dogs and cats hybridization...this has proved with some species only. Clarify or remove sentence.
L82-88: access to wildlife? Or allowing dogs roaming freely?

Methods
L92-99: Ethics should go at the end of methods
L116-126: This paragraph would fit better at the introduction where could add more emphasis on the need for conducting this study.
L135: CI of 95%?
L145-146: how degree of confinement was defined?
L166: what’s the aim of include rural and urban dogs as factors? Not clear the response variables on this analysis.
Definitions: How free-ranging dogs were defined?? How feral dogs are defined and how authors are sure they are not owned dogs? Just by interviews?
Statistic: aims is about dog and cats but the statistic is just regarding dogs, please clarify. Data on the questionnaire: it is not clear enough what authors wanted to test with the questionnaire. In methods it is not clear the method for model building and is confused with results section. M1: How a dog can be part of a free-roaming population? What is the response variable here? Dog restriction? M2, is regarding if a dog bring wild prey at home?? Model 3 not explained at all.
Discussion
There is data first given in the discussion and not informed in result section, such as distance of sighting dogs in wild area, among others.
Conclusion
How authors can be sure dogs and cats interact with wildlife?, only by interview?, there is any confirmatory method such as faecal analysis?

Tables and Figures
Figure 2. How sighting of dogs was defined??

---

## Round 0.2 · Minor Revisions

This manuscript has undergone substantial revision which has improved both content and clarity. Reviewer 1 now considers it suitable for publication. Reviewer 2 was not available, but a new reviewer (Reviewer 3) provided a detailed review. This reviewer considers the manuscript suitable for publication but has indicated some additional references and suggestions for clarification. My own reading of the manuscript identified a number of minor problems with word choice, grammar and punctuation which I have indicated on a pdf. In your rebuttal, please indicate any changes suggested on the pdf that you did not follow. In addition, I have one question regarding interpretation of your results. I am confident that the manuscript will be acceptable with these corrections and changes.

Editor's Comments
L444 It seems to me that hair in dog feces may not indicate predation but scavenging, for example from butchered carcasses that were available in a dump or elsewhere or from animals that died from other causes. Have you considered this interpretation?

Fig. 4. Caption refers to harassment but figure uses the term 'hunted', which is not necessarily the same and is inconsistent with the caption and text. I suggest replacing 'Hunted' with 'Harassed' on the figure.

Fig. 5. Caption refers to positive and negative association, but the traits are qualitative with no inherent positive and negative direction, so further explanation is required. You could possibly refer to Table 2, but you would need to be clear about the numerical values for each category which is not the present case for PREY, SEX, and LOCATION.

Reviewer 1 ·

Basic reporting

commented on the previous version

Experimental design

commented on the previous version

Validity of the findings

commented on the previous version

·

Basic reporting

The background and rationale for this research are described clearly, and the importance of exploring possible interactions between dogs, cats, people and wildlife in the Cape Horn Biosphere Reserve (CHBR) is made exlicit. In general, the standard of English expression is good (I have noted a small number of places for improvement below), and figures and tabular materials are clear. I have just a couple of suggestions to make on the basic reporting (in giving line numbers below I refer to the track-changed Word version, not the pdf), including a couple of references that may be relevant:

Line 54: The Cape Horn Biosphere Reserve is certainly a long way south, but it isn't strictly the southernmost protected area of the globe. That area would be Antarctica, a great deal of which is protected under international treaties.

Lines 74 onward: A recent reference that quantifies the global impacts of domestic dogs on wildlife is: Doherty et al. (2017 The global impacts of domestic dogs on threatened vertebrates. Biol Conservation 210, 56-59).

Conclusions: This section reads well and makes several well justified arguments about how the impacts of domestic dogs and cats might be reduced in the CHBR, based on the questionnaire survey results. A recent study by Hall et al. (2016, PLOS ONE 11(4) e151962) made similar recommendations, and more, based on questionnaire surveys carried out in six different countries, and should have some further recommendations of relevance here.

Experimental design

The questionnaire surveys and analyses of results appear to have been done well, and the results are presented clearly and fully. Importantly, the sampling and analytical designs allow the study's four main questions to be addressed and, with provision of the questionnaire, could be replicated by later researchers. The third aim could be clarified slightly. This says: "... examine experiences and perceptions of the impact of free-roaming dogs and cats on wildlife and corroborate those with an analysis of prey remains of dog feces". Why could clarification not be attempted with cat feces too? I presume these were too hard to find, but a short explanation would be helpful.

Validity of the findings

In general, I think the findings and conclusions are well made, with appropriate tempering of the initial results and conclusions that are drawn from them. Limitations are clearly identified and ways forward presented. This work will hopefully lead to actual quantification of the impacts of domestic dogs and cats on wildlife and other values in the CHBR, and to better mitigation of these impacts in future.

Additional comments

I was not a reviewer of the original manuscript, but my reading of the current version (with the track-changes) and of the authors' rebuttal letter suggests that it has been very extensively reworked and revised. It reads well, and most of the concerns that appear to have been present in the original version have been effectively addressed here. I noted just a small number of corrections that could be made to improve clarity or to improve the English expression:

Lines 101-102: "... a population of free-roaming dogs and cats ..." should be "... populations of free-roaming dogs and cats ...".

Lines 210-211: "(ii) report experienced problems associated with dogs ...". This is not clear; perhaps something like "(ii) report problems reported by respondents that were associated with dogs ..." better describes what is meant. Or, even more simply, just: "(ii) report problems associated with dogs ...".

Line 271: Were the 'street dogs' that were being fed on a regular basis owned or feral dogs?

Line 309: "Predominant experienced dog problems in town were conflicts ..." would read better as: "Predominant dog problems experienced in town were conflicts ...".

Line 470: Is 'provenience' meant to be 'provenance' or something similar?

Table 1: Please define what was meant by NAs.

Fig. 1: Yendegaia National Park is mentioned in the figure caption, but is not shown on the map. Can its location be added to the map?

---

## Round 0.3 · accepted · Accept

The changes made are appropriate and I consider the article now ready for publication.

# Note from PeerJ staff: We noted that you removed the text which refers to the supplemental material. Our apologies if our instructions were not clear, but it was not our intention for you to remove any text reference to the supplemental files. Our production group will work with you to re-insert that text. #